# Moving the Research Forward: The Best of British Biology Using the Tractable Model System *Dictyostelium discoideum*

**DOI:** 10.3390/cells10113036

**Published:** 2021-11-05

**Authors:** Robin S. B. Williams, Jonathan R. Chubb, Robert Insall, Jason S. King, Catherine J. Pears, Elinor Thompson, Cornelis J. Weijer

**Affiliations:** 1Centre for Biomedical Sciences, School of Biological Sciences, Royal Holloway University of London, Egham TW20 0EX, UK; 2UCL Laboratory for Molecular Cell Biology, University College London, Gower Street, London WC1E 6BT, UK; j.chubb@ucl.ac.uk; 3Institute of Cancer Sciences, University of Glasgow, Switchback Road, Glasgow G61 1QH, UK; Robert.Insall@glasgow.ac.uk; 4School of Biosciences, University of Sheffield, Firth Court, Western Bank, Sheffield S10 2TN, UK; jason.king@sheffield.ac.uk; 5Department of Biochemistry, University of Oxford, South Parks Road, Oxford OX1 3QU, UK; catherine.pears@bioch.ox.ac.uk; 6School of Science, University of Greenwich, Chatham Maritime, Chatham ME4 4TB, UK; E.Thompson@greenwich.ac.uk; 7Division of Cell and Developmental Biology, School of Life Sciences, University of Dundee, Dundee DD1 5EH, UK; c.j.weijer@dundee.ac.uk

**Keywords:** chemotaxis, *Dictyostelium discoideum*, slime mould, social amoeba, phagocytosis, autophagy, macropinocytosis, development

## Abstract

The social amoeba *Dictyostelium discoideum* provides an excellent model for research across a broad range of disciplines within biology. The organism diverged from the plant, yeast, fungi and animal kingdoms around 1 billion years ago but retains common aspects found in these kingdoms. *Dictyostelium* has a low level of genetic complexity and provides a range of molecular, cellular, biochemical and developmental biology experimental techniques, enabling multidisciplinary studies to be carried out in a wide range of areas, leading to research breakthroughs. Numerous laboratories within the United Kingdom employ *Dictyostelium* as their core research model. This review introduces *Dictyostelium* and then highlights research from several leading British research laboratories, covering their distinct areas of research, the benefits of using the model, and the breakthroughs that have arisen due to the use of *Dictyostelium* as a tractable model system.

## 1. Introduction

*Dictyostelium*-like species have been found in fossil records from 2.1 billion years ago [1], at the very beginning of multicellularity, positioning the Amoebozoa kingdom very early on in eukaryotic evolution. The family *Dictyostelia* is estimated to have evolved around 1 billion years ago [2]. Arising from this, the modern day *Dictyostelium discoideum*, which was originally called a ‘slime mould’, was initially used as a research model by Kenneth Raper in the 1930s [3,4,5] and then, later, by John Bonner [6,7], both of whom were fascinated with the development of this species as it transitions from a single celled ‘amoeboid’ form to a multicellular ‘fruiting body’. Over the following 90 years, research using *Dictyostelium* has provided fundamental insights into the biology of cell function and the process of multicellular development from a single cell to a mature fruiting body containing up to one million differentiated cells. It remains, by far, the best investigated model organism within the large group of the Amoebozoa.

The central reproductive cycle of *Dictyostelium* involves mitotic, single-cell division in the presence of nutrients. In nature, *Dictyostelium* cells are found in leaf litter in many habitats [8], where they consume microorganisms through a process called phagocytosis. The events leading to multicellularity are initiated at the onset of starvation (Figure 1), where an emerging process of collective cell signalling leads to a change in gene transcription and cell behaviour to promote multicellularity. Collective signalling guides directional cell movement, called chemotaxis, where cells both generate and sense a temporal gradient of cyclic AMP (cAMP), due to its secretion and the release of phosphodiesterase enzymes (PDEs) that degrade it [9]. Through this process, cells converge towards a dominant signalling centre (aggregation), to form ‘mounds’, within which cells continue to move in a spiralling pattern and initiate developmental changes into different cell types (primarily early spore and stalk cells). Further development of the mound leads to a ‘first-finger’ stage that can continue to develop through forming a ‘slug’ of around 1 mm in length. This slug is adapted to move towards light (phototaxis) [10], which is presumed to provide a means for the developing multicellular structure to rise above the leaf litter. Development continues, either directly from the finger stage or after slug formation, to form a mature fruiting body consisting of dead vacuolated stalk cells holding aloft a spore head (sorus) containing dormant and desiccation/starvation/temperature-resistant spores. Ultimately, spores released from the fruiting body germinate, renewing the cycle of single-cell division in the presence of nutrients.

Modern *Dictyostelium* research has flourished in a number of different directions due to a range of factors including the simple nature of the model, its distinct processes of growth and development, and a unique combination of research techniques. *Dictyostelium* represents a cellular model system with typical eukaryotic cell characteristics, including nuclei, Golgi apparatus, mitochondria and endoplasmic reticulum. The haploid genome of *Dictyostelium* has been fully characterised and annotated (dictybase.org) [11], including descriptions of each protein, phenotypes of mutants lacking individual proteins and related published material. The central focus of much research in this model remains in the areas of cell movement, communication and development. However, a range of other research areas have been well represented by *Dictyostelium* researchers including biomedical sciences [12,13] and neurodegeneration [14,15,16,17]; infection and immunology [18,19]; autophagy [20,21]; social cooperativity [22]; and cancer and DNA repair [23]. The reasons for using *Dictyostelium* as a research model derive from a range of key advantages over mammalian systems including. 

A simple system–A compact genome with reduced functional redundancy compared with mammalian cells, allowing clear analysis of mutant phenotypes;Developmental biology–Distinct lifecycles of single and multicellular phases, allowing the analysis of decisions specifically affecting cell proliferation and differentiation;Biochemistry–The rapid and inexpensive production of multiple grams of isogenic cells, for example, to analyse enzyme activity, metabolic changes, organelle function or development;Genetics and cell biology–A genome wide *Dictyostelium* mutant library, providing the ability to screen the genome for gain or loss of function phenotypes [24];–Rapid (multiple) gene ablation or overexpression in isogenic cell lines to identify and characterise protein function (including human proteins) [25];Microscopy–Advanced microscopy techniques to visualize, monitor and analyse protein and RNA dynamics;Pharmacology–Pharmacogenetic approaches, where screening of mutant-cell libraries can identify the genetic basis of cell or developmental behaviour or how drugs or related compounds act in cells (i.e., mechanism of action studies);–The rapid analysis of compound effects on normal cell function, including acute and chronic exposure and structure/activity studies.

In this review, we highlight some of the world-leading research in the field of *Dictyostelium* biology arising from UK-based researchers. These research excerpts are used to introduce the various areas that are being investigated using this model, show how *Dictyostelium* provides an advantageous system for research, detail the arising discoveries, and demonstrate the wider impact of these studies.

## 2. Understanding Cell Fate Decisions

Despite intense activity in the field of developmental biology for many decades, our understanding of how cells make decisions about their fate is decidedly limited. Models tend to fall into two categories—instructive or permissive. Instructive models emphasise the external or inherited trigger, giving the cell no choice. Permissive models view the cell as waiting for some event that triggers a decision that has already been made, often incorporating a central “noise plus feedback” element. These models can be useful in the limited number of apparently simple contexts in which cell fate appears to be determined by a limited number of components. However, in most systems, which are neither wholly instructive nor permissive and often depend on a highly complex set of interactions between the cell and its environment [26], these opposing models lack utility.

Although progress has been made by studying specific molecules, these approaches do not reveal the complexity of the niche that interacts with the cell, and we are far from having a fully parameterised catalogue of all the salient molecular interactions occurring both inside and outside the cell. The primary problem we face is determining the appropriate level of organisation at which we should focus our efforts and measuring all of the implicated agents in the decision-making process (such as signals, cells, transcription factors, etc.) within a meaningful physiological dynamic range. Ideally all of the major signalling inputs and cellular responses should be quantified in real time in the physiological niche. These measurements require monitoring of the dynamics, strength and distribution of signals, together with multiple real-time readouts of cell states, such as transcription, motility and proliferation. 

These approaches are challenging to bring together in vivo; however they can be combined in *Dictyostelium*. This is a leading system for understanding signalling in differentiating cell populations, as the signalling machinery is highly conserved in mammalian cells [27]. The decision to differentiate in *Dictyostelium* is accessible to noninvasive imaging from submicron to millimetre length scales [28], so it is possible to directly monitor the entire heterogeneous decision-making process within individual cells in the context of the collective. To define the signalling history of a cell, the dynamics of the major extracellular signals can be quantified in relation to cell positions in the niche. In addition, these signals are amenable to precise genetic and optogenetic manipulation. Crucially, the transcriptional outputs underlying decision making can be directly measured in real time using technology developed in *Dictyostelium* to image transcription dynamics in single cells [29,30,31,32] (Figure 2). 

## 3. Self-Generated Chemotactic Gradients

Directed cell movement in response to chemical gradients, or chemotaxis, is used by many cells, including *Dictyostelium*, cancer cells of different origins (especially pancreatic cancer and melanoma) and immune cells such as neutrophils. In this scenario, *Dictyostelium* acts as a gateway organism; its simple genome, ease of culture and our huge library of mutants (both in our own freezers [33] and through Dictybase [34]) make it an ideal subject for proof-of-principle experiments.

Most research laboratories that study chemotaxis start from the basis that someone else creates the gradient, and the chemotactic cell’s sole job is to interpret it [35,36,37]. This is unlikely to occur very often in real physiology. Cells can obtain more information and thus respond more strongly by breaking down attractants while they move [38]. In a more exaggerated situation, cells can create their own gradients, even when none are initially present in the environment. If an attractant is plentiful, cells are localized, and they break down the attractant. Then, gradients of some sort are inevitable, with the highest density of cells corresponding to the low end of the gradient. These are known as ‘self-generated’ gradients [39], and *Dictyostelium* is one of the best organisms for studying them.

Several features of *Dictyostelium* biology make it ideal. The attractants are well understood, and chemical analogues with different properties are available, for example, attractants that cannot be broken down but which still trigger cell movement. Cyclic AMP (cAMP) is the most-studied attractant and can be mimicked by the non-hydrolysable molecule Sp-cAMPS, where cell responses are strikingly different [40]. *Dictyostelium* uses cell-surface enzymes to break down its favoured chemoattractants. cAMP is degraded by cAMP phosphodiesterase (PdsA) [41], and folate secreted by bacteria is broken down by folate deaminase, which has been identified biochemically [42] but is yet to be cloned. Sp-cAMPS is not altered by PdsA, and similarly, methotrexate is unaffected by folate deaminase. The combination of receptor mutants, migration mutants, chemoattractant analogues and a vibrant research community has allowed us to define self-generated gradients better in *Dictyostelium* than anywhere else.

The most arresting demonstration of the power of self-generated gradients is shown by cells that can solve mazes [43]. We built mazes of different complexities using the same process as was used in the *Dictyostelium* Cell Race [44]—PDMS silicone rubber cast on etched silicon blanks. Using these, it was shown that cells can make well-informed decisions about the best path they can take. Without seeing the path or any a priori gradients or information, the self-generation probes the environment and allows the cells to discover where they should go before they arrive at decision points. Many other cells use the same process, but *Dictyostelium* has been the organism of choice to understand it.

## 4. Macropinocytosis

Another area where research in *Dictyostelium* has made a significant impact is the study of macropinocytosis. This is the process by which cells engulf large volumes of extracellular fluid via the extension of large cup-shaped protrusions that close and collapse in upon themselves. This causes the formation of a large intracellular vesicle containing extracellular proteins and macromolecules which immediately begins to be processed and digested within the cell. 

Whilst macropinocytosis has been subverted by vertebrate immune cells such as macrophages and dendritic cells to survey the environment and capture antigens for presentation, the original evolutionary role for drinking extracellular fluid as food is most likely to feed [45]. This is exemplified in *Dictyostelium* and other protists, which can use macropinocytosis as an alternative to phagocytic feeding, allowing them to grow in liquid medium (Figure 3). Importantly, this has been conserved all the way to human cells, and although most unstimulated cells do not take up much fluid, macropinocytosis is highly upregulated in many cancers, endowing them with the ability to eat extracellular proteins to sustain their unrestrained growth [46]. This, and the finding that macropinocytosis acts as an entry mechanism for bacterial and viral pathogens as well as prion aggregates and even biotherapeutics such as RNA vaccines has led to a surge of interest in macropinocytosis in recent years [47]. 

*Dictyostelium* has proven to be an ideal model system for investigating the organisation and dynamics of macropinocytic cups as well as determining how the resultant vesicles are processed. Key to this are the large and frequent cups formed by these cells. When combined with high resolution microscopy, this allows macropinocytosis to be observed in greater detail than in any other system. With the added advantages of simple genetic manipulation and the ability to perform genetic screens, *Dictyostelium* provides a unique opportunity to understand the basic and highly conserved mechanisms of macropinocytosis. 

This has allowed major advances, such as the proposal of a “template” model for macropinocytic cup formation, where the cup shape is formed by organising a ring of actin polymerisation around a central domain of the signalling lipid PIP_3_ [48]. *Dictyostelium* studies have also led the way to understanding how the Ras and Rac small GTPases combine to regulate cup size and dynamics [49,50] as well as illuminating how the cytoskeleton is differentially regulated across the cup [51,52]. Large cups make large vesicles, which are, again, highly tractable to imaging and can be followed over time. This allows intracellular processing to be observed directly, leading to discoveries, such as how surface proteins are retrieved from macropinosomes to save them from degradation [53]. 

Whilst interesting and important themselves, studies of macropinocytosis have an additional impact beyond fluid uptake. Macropinocytosis shares its evolutionary origins and nearly all of its engulfment and processing machinery with bacteria feeding by phagocytosis. Macropinocytosis studies, therefore, often also have implications for phagocytosis, which can easily be studied in parallel in *Dictyostelium* [49,50,51,53]. As a professional phagocyte, *Dictyostelium* has been used extensively as a model host to study interactions with a wide range of both bacterial and fungal pathogens [19,54]. This adds significant value to studies of the core macropinocytosis and phagocytosis machinery. 

Recent years have seen a rapid increase our understanding of macropinocytosis. *Dictyostelium* research has been at the forefront throughout, proving to be a useful and highly conserved model. Many questions still remain, such as the mechanism by which actin polymerisation is enticed into a ring, how the cups close, how the cell recognises a newly formed macropinosome and how maturation is regulated. It, therefore, seems likely that research using *Dictyostelium* will remain important and contribute further information for some time to come.

## 5. Targeting Histone Post-Translational Modifications

Post-translational modifications of histone proteins play a vital role in regulating the interaction of nucleosomes with proteins involved in a range of processes, including gene expression and DNA repair. A large number of modifications have been described in mammalian cells with combinations adding to the complexity. Histone modifications are frequently aberrant in cancer cells, and a number of pharmacological inhibitors of modifying enzymes specifically target cancer cells. *Dictyostelium* offers a unique system to study the mechanistic function of histone modifications in a system with multicellular complexity. Modifications are highly conserved, including many that are absent in models such as yeast. Importantly the genome contains mainly single copies of genes encoding histone variants, unlike mammalian cells which can contain more than ten gene copies, precluding genetic manipulation [55]. This facilitates rapid generation of *Dictyostelium* strains with mutations introduced into the endogenous genes to generate versions of histones that lack individual modification sites [56,57]. There is an emerging role of the nuclear lamina in epigenetic regulation through histone modifying enzymes. Since many nuclear lamina components of mammalian cells are conserved in *Dictyostelium*, this makes the system even more attractive to study post-translational histone modifications. *Dictyostelium* also shows reduced redundancy in modifying enzymes, often having a single orthologue [58], which again facilitates the generation of strains lacking specific modifications and genetic screens to characterise their role as well as identify drug-resistant mutants. Thus, *Dictyostelium* can be used to understand the mechanism of action and modes of resistance against inhibitors which impact histone modification and are currently in use in the clinic for cancer therapy. 

Histones associated with active genes are acetylated and deacetylase inhibitors (KDACi) that specifically induce the apoptosis or differentiation of cancer cells. In mammalian cells, Histone 3 Lysine 4 (H3K4) trimethylated histones, enriched at promoters, are the most rapidly acetylated during treatment of cells with KDACi, revealing dynamic turnover of acetylation on these histones [59]. This link is conserved in *Dictyostelium* [60], and KDACis inhibits development, allowing the importance of this process in the mechanism of action of KDACi to be determined [61]. Strains that are deficient in the only H3K4-methyltransferase gene show resistance to KDACi inhibition of development. Mutation of the methylation site in the endogenous H3 variant genes generates strains lacking the dynamic pool of acetylated histones and that also show resistance. This reveals the importance of this pool of dynamically acetylated histones in the mode of action of these drugs. Gene disruption studies have revealed that a subunit of an acetyltransferase that binds H3K4me is required for this link, suggesting that drugs that target this binding will provide an alternative to KDACi. Many cancer cells show overexpression of this protein, providing a potential biomarker for sensitivity to this class of drugs.

A distinct histone modification that is targeted in cancer treatment is ADP-ribosylation. DNA damage, such as double strand breaks, can lead to genome instability and cancer. ADP-ribosyltransferases (PARPs) trigger ADP-ribosylation of histones at damage sites, but the mechanistic consequences of this are not known. PARP inhibitors are currently used to treat breast and ovarian cancers that are deficient in one pathway of double strand break repair [62]. DNA damage-responsive ADP-ribosyltransferases are conserved in *Dictyostelium*, although they are missing in other model organisms [23], and characterisation of strains deficient in single or multiple related enzymes have revealed redundant roles in repair of different types of DNA damage [63,64]. This has important implications for the use of specific inhibitors for individual PARPs in cancer cells with defects in different response pathways. Multiple histones are modified at different sites, so strains with endogenous histone proteins lacking combinations of ADP-ribosylation sites are being generated to reveal their importance in genome stability [65].

Studies on *Dictyostelium* are, therefore, providing unique insights into the importance and mechanisms of action of these histone modifications and their roles in determining the sensitivity of cancer cells to these chemotherapeutic drugs.

## 6. Evolutionary Membrane Biology 

Despite the ancient origins of the Amoebozoa, the proteome of the amoeba *Dictyostelium* remains remarkably similar to that of animals, providing an opportunity to compare functions of evolutionarily early but undescribed proteins in membrane biology. This is particularly relevant for the conservation of mitochondrial proteins and functions, established areas of *Dictyostelium* research [14], including one of the most recently discovered protease types, the ‘rhomboid’ proteases. These enzymes operate in the hydrophobic environment of the membrane, often providing a regulatory switch as they cleave their substrates. They can be described as being necessary for life, since they are ubiquitous across evolution. Recently, these proteins have been suggested to be involved in human disease as well as providing an interesting paradigm for signalling and regulation from the membrane. In addition, work on organellar rhomboids of plants [66,67] showed roles in membrane lipid composition for plastid rhomboid orthologues with pleiotropic effects that extend to floral morphology. Regulation of the lipid content of membranes may be an important conserved function of rhomboids, since a mitochondrial representative was also shown to cleave a lipid transfer protein, governing the dimerisation of ATP synthase and cristae morphology [68]. In addition to studying the four active rhomboids of *Dictyostelium* [69], this similarity in mitochondrial regulation across organisms has enabled its use as a model to investigate how mitochondria might be treated to reduce age-related macular degeneration [70].

The parallels between plant and amoeba membrane biology have also inspired work on a group of membrane transporter proteins. Multidrug and toxin efflux (MATE) transporters are encoded by large multigene families in plants, operating in ion and pigment transport at the vacuole [66]. The smaller, less selective groups in bacteria are a key cause of antibiotic resistance. As MATE proteins are also found in human cell membranes, they can contribute to drug resistance in human anticancer therapy. *Dictyostelium*, again, closely parallels human cells, with a pair of MATE proteins whose roles had not previously been documented. Ongoing research in this area is exploring the roles of MATE proteins, including their substrate specificity, physiological relevance (for example in phagocytosis of prey bacteria), and the parallels in gene expression between the amoeba transporters and those in the human kidneys and liver [71]. 

Thus, even though *Dictyostelium* is a relatively simple organism, it provides a route for understanding the modes of action of new drug treatments. Using flavonoids as a model drug class, applied research includes investigation of the ability of *Dictyostelium* to efflux compounds during assays of activity and the repurposing of a plant reporter for use as a simple but quantifiable indicator of the entry and efflux of polyphenolic compounds [71].

## 7. Cellular Mechanisms Underlying Multicellular Morphogenesis

A major goal in developmental studies is to understand the principles that underlie the integration of critical cellular behaviours by cell signalling to drive multicellular development and morphogenesis. This has been investigated in a wide variety of organisms ranging from very simple to very complex. *Dictyostelium* is a simple organism found at the boundary between unicellular and multicellular organisms in the tree of life. The developmental cycle has been studied extensively, since development takes only 24 h and is a simple system as there are only a few cell types and the growth and developmental phases are separated in different parts of the lifecycle [72]. The multicellular developmental cycle is triggered by starvation and occurs essentially in the absence of cell division, but the time of exit biases the differentiation of the cells into stalks or spores [73]. The various multicellular stages, aggregates, mounds, migrating slugs and fruiting bodies are formed through the rearrangement of differentiating cells in space and time.

3’-5’cyclic AMP (cAMP), the chemoattractant that controls the aggregation phase of development, was one of the first chemoattractants to be identified [74]. Triggered by starvation, cAMP starts to be produced and secreted by the aggregating cells in a periodic manner. Release of cAMP starts an autocatalytic amplification cycle, resulting in rapid secretion of cAMP. This secreted cAMP is detected and amplified and secreted by surrounding cells to propagate as spiral and concentric waves of chemoattractant through the population of cells [27,75]. These chemoattractant waves propagate from aggregation centres, where they are periodically initiated outward and direct the inwards chemotactic movement of the cells towards the aggregation centre. *Dictyostelium* can be used to study the dynamics of the cell–cell signalling mechanisms that control the movement behaviours of the differentiating cells and the feedback of these cell movements through cell signalling using a combination of experimental and modelling approaches. We have shown that waves of chemoattractant organise cell movement not only during aggregation but also in the later multicellular stages of development. We initially developed methods to visualise the waves of chemoattractant via light scattering changes associated with periodic cell movements during aggregation in mounds and in migrating slugs, followed by visualisation of changes in phosphatidyl-inositide phosphate signalling associated with cAMP mediated signal transduction, as reviewed in [76]. More recently, we have been able to monitor the dynamics of cAMP signalling in vivo during all stages of the lifecycle using cAMP specific FRET constructs (Figure 4) while simultaneously characterising the characteristics of the chemotactic cell movements [77].

Due to the visualisation of both the dynamics of the chemotactic signals and the resulting chemotactic response and interactions between them during all stages of development, it has been possible to classify *Dictyostelium* morphogenesis as a highly self-organised process based on two essential cell behaviours: excitable cell–cell signalling and chemotaxis, resulting in robust multicellular morphogenesis [78]. Many questions remain to be resolved regarding the detailed molecular mechanisms associated with cell polarisation and cytoskeletal dynamics during the chemotactic response and the role of cell contact in cell polarisation in the later stages of development [79,80,81,82]. 

## 8. Molecular Medicine and Drug Discovery

Epilepsy is one of the most common serious neurological conditions worldwide. Despite a century of research into the treatment of this chronic condition, around one-third of patients continue to experience seizures, even though current medicines available [83]. Thus, there remains a clear and urgent need to develop better treatments for these individuals. The origins of seizures is largely considered to be a slight imbalance in the excitation or inhibition of neuronal signalling, giving rise to localised or generalised hyper-activation of regions of the brain and a wide range of behavioural responses defined as seizures [84]. Due to these mechanisms, the vast majority of research into epilepsy and novel treatments focus on synaptic signalling and employ animal models, and a wide range of compounds have shown efficacy in blocking seizure-like activity. How these drugs work at the molecular and cellular levels has often been proposed but rarely agreed on. 

Research using *Dictyostelium* as a simple biomedical model has provided an innovative approach to investigate molecular mechanisms for epilepsy treatments, and from this, the development of potential new drugs that have been shown to block seizures has occurred. This model provides the ability to assess the bioactivity of one of the most widely used treatments for epilepsy and bipolar disorder, valproate (valproic acid; VPA) [85], a short-chain branched fatty acid (2-propylpentanoic acid). This drug provides excellent broad-spectrum seizure control but is likely to be withdrawn from widespread use due to its significant side effect of teratogenicity [86,87]. To identify the molecular mechanism of valproate using *Dictyostelium,* initial studies focused on developmental effects to show that physiologically relevant concentrations of valproate block fruiting body formation, suggesting a specific molecular effect related to cell signalling in development [88]. To investigate this mechanism, and since early development is regulated by cAMP-induced chemotaxis, leading to the formation of phosphoinositides, further studies found that valproate treatment reduces phosphoinositide signalling (formation of phosphatidylinositol phosphate [PIP] and phosphatidylinositol 4,5-bisphosphate [PIP_2_]) (Figure 5), a mechanism that had not been identified in other models [89]. The effect on phosphoinositide turnover was then confirmed using in vitro and in vivo analyses in rodent models, and the results suggested that a similar effect is likely in humans, potentially related to epilepsy treatment [90]. Identification of this mechanism then enabled a low-throughput drug screen to identify related compounds with improved potency. Here, over 100 related compounds were assessed for the ability to reduce phosphoinositol signalling [89] to identify a range of structurally specific medium-chain fatty acids with potent activity in the reduction of phosphoinositide signalling. The relevance of these compounds for epilepsy treatment was then confirmed, and many of them were found to provide more potent seizure control in ex vivo mammalian seizure models [89,91,92]. In collaboration with the US Epilepsy Therapy Screening program, several of these compounds were also shown to block seizure activity in a range of animal models representing different types of epilepsy, including for models of generalized tonic-clonic seizures, focal seizures and absence seizures [92]. Interestingly, one of the compounds showing potent seizure control was decanoic acid, which is a key component of a diet used to treat patients with drug-resistant epilepsy [93]. Further analysis of decanoic acid identified multiple antiepileptic mechanisms, which were validated in mammalian models, including direct inhibition of AMPA receptors [94] and inhibition of mTORC1 [95]. Excitingly, several of the compounds identified, including trans-4-butylcyclohexane carboxylic acid (4BCCA) and 4-ethyloctanoiac acid (4EOA) [92], showed improved potency over valproate in multiple ex vivo and in vivo models and are likely to lack teratogenic side effects, providing a promising new medicinal approach for the treatment of patients with generalised and drug-resistant epilepsy.

Thus, through using *Dictyostelium* as a pharmacogenetic model to identify mechanism(s) of action of commonly used medicines, it is possible to translate discoveries to preclinical models and identify more potent drugs with reduced side effects. 

## 9. Conclusions

The use of *Dictyostelium* as a tractable model for biological research has progressed from initial studies regarding the unusual process of cell aggregation and development shown by the model, to now include a wide array of research areas. British research centres have focused on understanding how future changes in cell fate are determined, how cells function in cell-to-cell signalling, how cells consume extracellular liquid, how cells regulate gene expression and DNA repair, fundamental aspects of cell membranes and proteins within membranes, processes by which cells change structure and function in development, and investigations of the mechanisms of action of medicines and bioactive natural products. Underlying all these areas of study is the relative simplicity of the system (at both a genetic level and in development) and the range of specific research approaches that are difficult to use in other research models. In addition, the close-knit and collaborative research community and support by the international resource for the model system (Dictybase) provides a strong and vibrant platform for innovative and translational research. Thus, *Dictyostelium* provides an excellent research model with the ability to significantly impact a wide range of fields within biology.

## Figures and Tables

**Figure 1 cells-10-03036-f001:**
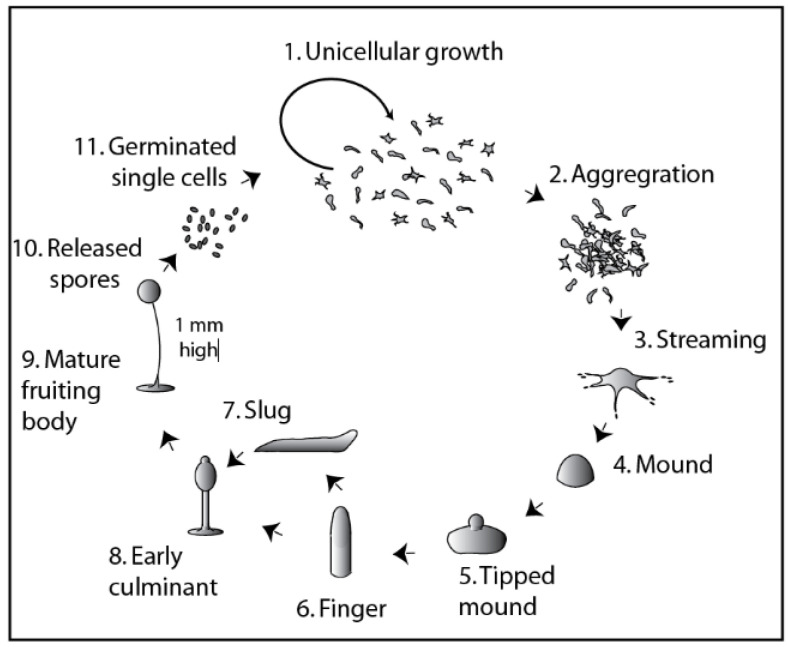
*Dictyostelium discoideum* has both single-celled and multicellular phases in its life cycle. *Dictyostelium* cells proliferate in the unicellular phase by mitotic division. Upon starvation, cells aggregate through chemotaxis to cAMP. Development proceeds through streaming, mound and tipped mound formation, leading to the multicellular finger stage. At this stage, development can bifurcate, either forming a migratory slug-like structure or progressing straight to an early culminant and finally a mature fruiting body of around 1 mm in height, containing spores in a spherical spore head. These dormant cells are resistant to dehydration and starvation and, subsequently, are released to renew their cycle of unicellular growth.

**Figure 2 cells-10-03036-f002:**
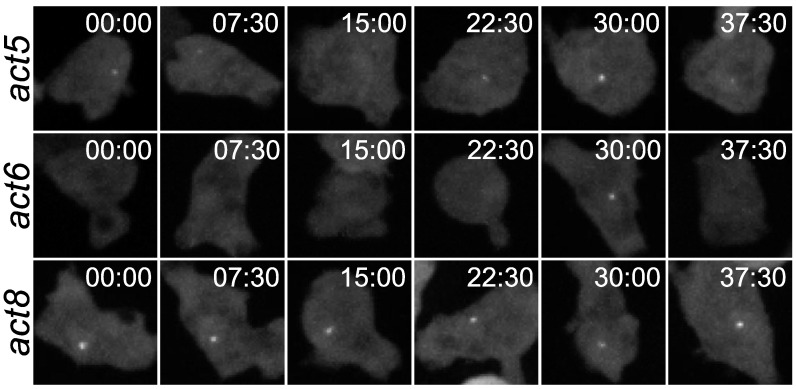
Imaging transcription in living cells. Images are stills from movies of *Dictyostelium* cells with different actin genes tagged using the MS2 RNA detection system. Nascent RNA appears at the site of transcription as a fluorescent spot. Different genes show different spot appearance and disappearance kinetics, with act5 and act6 genes being more “bursty” and act8 more constitutive. Image reproduced from [31]. Bursting transcription is now known to be conserved in all forms of life, from bacteria to mammalian cells, and has been implicated as a driver of cellular heterogeneity in diverse contexts- such as underlying HIV latency, the persistence of bacteria and cancer cells exposed to drug treatment, and cell fate separation during development.

**Figure 3 cells-10-03036-f003:**
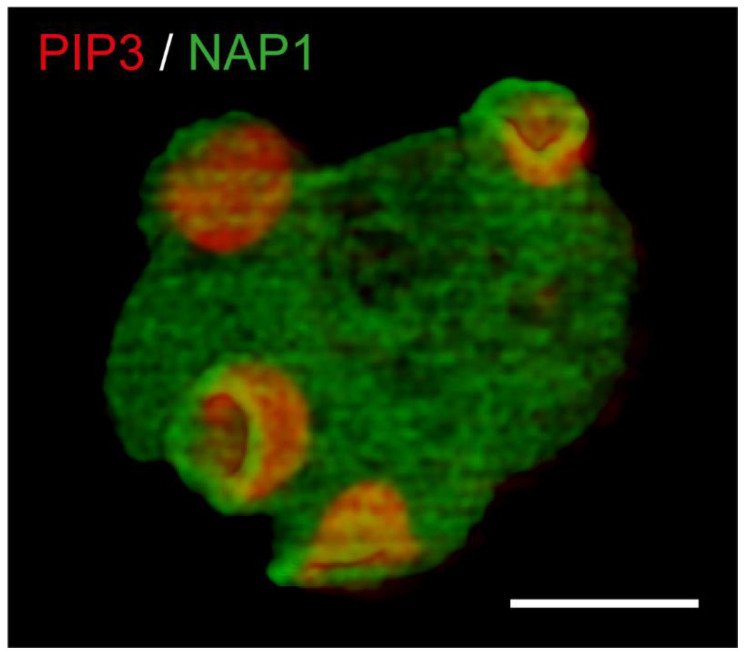
Macropinocytic cup formation in *Dictyostelium*. A 3D projection from a timelapse of *Dictyostelium* cells co-expressing a reporter for the signalling lipid PIP_3_ (PH-pkgE-RFP) that designates the cup interior and is encircled by a ring of the SCAR/WAVE complex (shown by NAP1-GFP). Images taken by lattice light sheet microscopy. Bar = 10 μM.

**Figure 4 cells-10-03036-f004:**
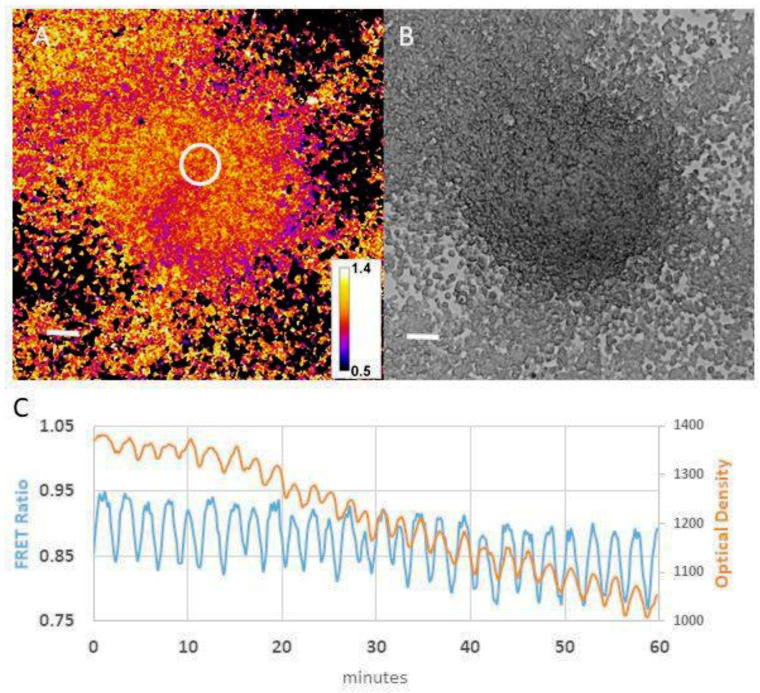
Comparison of cAMP and optical density waves during late aggregation. (**A**) Image showing the dynamics of cAMP waves in a streaming aggregate. Intracellular cAMP variations were measured using a cAMP specific FRET sensor. The sensor is based on the mammalian EPAC cAMP binding protein, and cAMP variations are measured as donor-acceptor emission ratios [77]. (**B**) Optical density image of the same aggregate as shown in (**A**). (**C**) FRET ratio and optical density changes as functions of time measured in the circle region shown in (**A**).

**Figure 5 cells-10-03036-f005:**
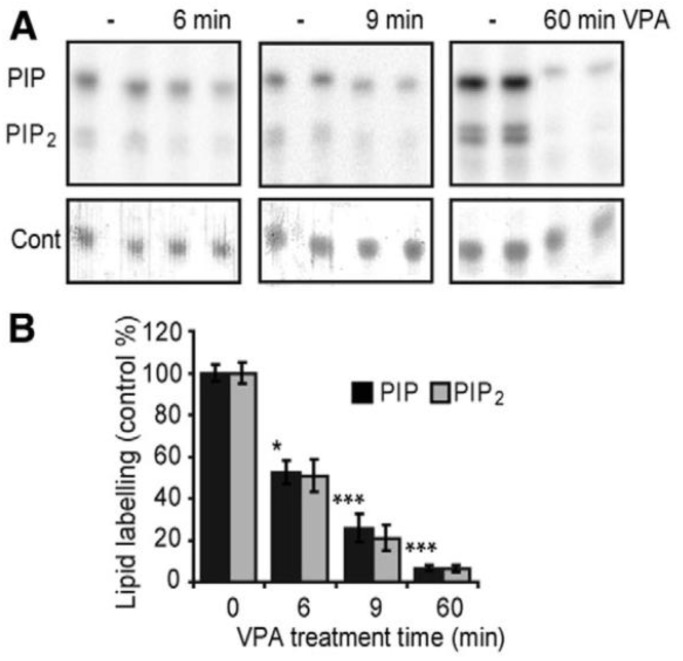
Identifying a molecular mechanism associated with valproate that acts on *Dictyostelium* phosphoinositide signalling. (**A**) *Dictyostelium* cells, induced to the early aggregation stage, were permeabilized in the presence of radio-labelled ATP and phosphatase inhibitors for 6, 9 or 60 min in the absence (-) or presence of 0.5 mM valproate (VPA). (**B**) Levels of PIP and PIP_2_ were quantified in comparison to total lipids (Cont). This assay identified, for the first time, that valproate reduces phosphoinositide signalling and was subsequently used to identify a range of second-generation valproate replacements that showed strong seizure control in pre-clinical models. * *p* < 0.05; *** *p* < 0.001 for PIP levels, from [89].

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
