# Peer review of "Moving the Research Forward: The Best of British Biology Using the Tractable Model System Dictyostelium discoideum"

_cells, 2021, doi:10.3390/cells10113036_

Round 1

Reviewer 1 Report

With their manuscript Williams and co-authors present a showcase review on Dictyostelium as a model system. The focus on Britain can be explained as this is an invited review for a special issue of Cells focusing on "State-of-the-Art and Perspectives in the British Isles". Despite I feel that focusing such a review on a certain country and not only on the subject itself is not ideal, I do not want to question the concept of the special issue here.

All in all, this is a manuscript worth reading and I have only a few remarks that should be addressed in a revised version that I would be happy to see published in this journal.

  • line 66: spores are also temperature resistant.

  • I think it may be useful to emphasize that Dictyostelium is by far the best investigated model organism within the large group of the Amoebozoa.

  • Line 276: the last sentence of the chapter is a bit too weak. I would suggest either to delete it or to emphasize the impact of Dicty for this research more clearly in this last sentence.

  • Line 278: there is an emerging role of the nuclear lamina in epigenetic regulation through histone modifying enzymes. Since many nuclear lamina components of mammalian cells are conserved in Dictyostelium, this makes the system even more attractive to study post-translational histone modifications.

  • Line 371. the introductory sentences are a bit redundant as the position in the tree of life and development was already discussed before.

  • Line 407: For outsiders in the field Fig. 4 is not really helpful in the way it is presented here. The legend is too short. The experiment should be explained in a way that it is not necessary to read the cited papers. Which fluorophores were used, what are the FRET partners, what kind of setup was used?

Author Response

Reviewer 1: With their manuscript Williams and co-authors present a showcase review on Dictyostelium as a model system. The focus on Britain can be explained as this is an invited review for a special issue of Cells focusing on "State-of-the-Art and Perspectives in the British Isles". Despite I feel that focusing such a review on a certain country and not only on the subject itself is not ideal, I do not want to question the concept of the special issue here.

We understand and agree with the reviewers comments here, but never-the-less, have taken this opportunity to support research in this areas.

All in all, this is a manuscript worth reading and I have only a few remarks that should be addressed in a revised version that I would be happy to see published in this journal.

We thank the reviewer for these positive comments.

  • line 66: spores are also temperature resistant.

We have added this to the sentence

  • I think it may be useful to emphasize that Dictyostelium is by far the best investigated model organism within the large group of the Amoebozoa.

This is a very good suggestion, and we have added this to the end of the first paragraph.

  • Line 276: the last sentence of the chapter is a bit too weak. I would suggest either to delete it or to emphasize the impact of Dicty for this research more clearly in this last sentence.

We agree (it is a bit corny). We have added ‘Thus Dictyostelium provides excellent research model with significant impact in a wide range of fields within biology’.

  • Line 278: there is an emerging role of the nuclear lamina in epigenetic regulation through histone modifying enzymes. Since many nuclear lamina components of mammalian cells are conserved in Dictyostelium, this makes the system even more attractive to study post-translational histone modifications.

This has been corrected as suggested.

  • Line 371. the introductory sentences are a bit redundant as the position in the tree of life and development was already discussed before.

This has been corrected in the manuscript (shown with track changes)

  • Line 407: For outsiders in the field Fig. 4 is not really helpful in the way it is presented here. The legend is too short. The experiment should be explained in a way that it is not necessary to read the cited papers. Which fluorophores were used, what are the FRET partners, what kind of setup was used?

We have corrected the legend as requested.

Reviewer 2 Report

Due to the objectives of the special issue, this review is covering most  of the areas of research of different British labs working on Dictyostelium. These labs are not collaborating together on a single topics, but each one is working on different questions sometimes unrelated to each other. What they have in common is the same model organism, Dictyostelium, which due to its life cycle and its amenability to molecular genetics and cell biological approaches, can be used to tackle different questions.  As the authors state in the introduction: "in this review we highlight some of the world-leading research in the field of Dictyostelium biology arising from UK-based researchers”. 

Some of the topics are relevant and interesting, such as “How cells undergo different patterns of differentiation during development” “How self-generated gradients are formed in chemotaxis”; “how macropinocytic cups are formed and regulated”; “Which post-translational histone modifications are conserved in Dictyostelium and how sensitive is Dictyostelium to chemotherapeutic drugs targeting these modifications”; “How Dictyostelium has been used to study the regulation of mitochondrial membrane lipids by rhomboid proteases and the role in plasma membrane transport of two newly identified multi-drug and toxin-efflux transporters that are shared by Dictyostelium and mammalian cells”; “Which are the mechanisms regulating multicellular morphogenesis”; “How Dictyostelium has been used as biomedical model to decipher the molecular mechanisms underlying epilepsy and as  apharmacogenetic model for the discovery of  new drugs”.

Some of these topics are very original, such as those on chemotaxis or macropinocytosis. 

I confirm that the paper is well written, clear and easy to read.

Author Response

Reviewer 2

Due to the objectives of the special issue, this review is covering most of the areas of research of different British labs working on Dictyostelium. These labs are not collaborating together on a single topics, but each one is working on different questions sometimes unrelated to each other. What they have in common is the same model organism, Dictyostelium, which due to its life cycle and its amenability to molecular genetics and cell biological approaches, can be used to tackle different questions.  As the authors state in the introduction: "in this review we highlight some of the world-leading research in the field of Dictyostelium biology arising from UK-based researchers”. 

Some of the topics are relevant and interesting, such as “How cells undergo different patterns of differentiation during development” “How self-generated gradients are formed in chemotaxis”; “how macropinocytic cups are formed and regulated”; “Which post-translational histone modifications are conserved in Dictyostelium and how sensitive is Dictyostelium to chemotherapeutic drugs targeting these modifications”; “How Dictyostelium has been used to study the regulation of mitochondrial membrane lipids by rhomboid proteases and the role in plasma membrane transport of two newly identified multi-drug and toxin-efflux transporters that are shared by Dictyostelium and mammalian cells”; “Which are the mechanisms regulating multicellular morphogenesis”; “How Dictyostelium has been used as biomedical model to decipher the molecular mechanisms underlying epilepsy and as  apharmacogenetic model for the discovery of  new drugs”.

Some of these topics are very original, such as those on chemotaxis or macropinocytosis. 

I confirm that the paper is well written, clear and easy to read.

  • We thank the reviewer for these positive comments.